# Slot Design Optimization for Copper Losses Reduction in Electric Machines for High Speed Applications

**Claudio Bianchini** *[ID], **Mattia Vogni, Ambra Torreggiani**[ID]**, Stefano Nuzzo**[ID]**, Davide Barater**[ID] **and Giovanni Franceschini**

DIEF, University of Modena and Reggio Emilia Italy, 41121 Modena, Italy; 240925@studenti.unimore.it (M.V.); ambra.torreggiani@unimore.it (A.T.); stefano.nuzzo@unimore.it (S.N.); davide.barater@unimore.it (D.B.); giovanni.franceschini@unimore.it (G.F.)
* Correspondence: claudio.bianchini@unimore.it

**Abstract:** The need of a wide operating range and a high power density in electric machines for full- and hybrid electric vehicles in traction applications has led to an increase in the operating frequency of the machine. When the electric frequency increases, the additional losses in stator windings become an issue and they have to be taken into account in the design of the electric machine. This issue is more critical when hairpin windings are employed, due to the the skin and proximity effects which produce increased copper losses. In this paper, the relationships between different stator slot parameters (tooth width, slot opening, etc.) and stator winding copper losses have been analysed in order to identify an optimal design of a single stator slot .

**Keywords:** hairpins winding; skin effect; proximity effect; high speed drive

## 1. Introduction

Nowadays, it is common for traction machines in battery-electric vehicles (BEV) and hybrid electric vehicles (HEV) to operate with a high phase current (hundreds of Amps). Therefore, hairpin windings are usually chosen for electric machines in traction application [1]. Differently from the traditional windings, hairpins are formed by rectangular bars [2], which are bent into a hairpin shape [3], then they are axially inserted inside the stator slot and terminals are finally bent and welded with the terminals of the other coils [4]. Figure 1 shows an elementary hairpin, while Figure 2 shows a scheme of the manufacturing process of the winding. The authors of [3–5] offer more details of the hairpin winding manufacturing process.

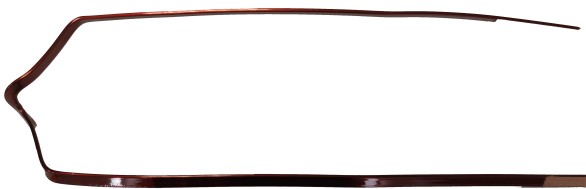

**Figure 1.** Elementary hairpin.

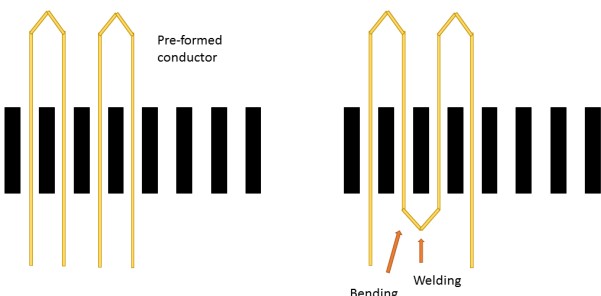

**Figure 2.** Manufacturing process of hairpin winding.

The rectangular cross-section area of hairpins leads to a higher slot-filling factor ($k_{fill}$) and a more rational end windings design, that yields a high reduction in the Joule losses when the electric machine needs to operate at high torque and low speed. The hairpin windings increase the conductor ampacity and allow an easier heat transfer from the slot to the stator lamination.

On the other hand, even if hairpin windings guarantee a higher efficiency at high torque and low speed, their higher cross-section area reduces the benefits of a greater $k_{fill}$ when the electric machine operates at high speeds. This effect is due to two high-frequency phenomena: the *skin effect* and the *proximity effect*. The current density distribution becomes uneven at high operating frequencies, leading to an increase in the effective resistance and, hence, of the Joule losses.

Nowadays, the request for high performance in BEV and HEV leads to an increase in the electric machine phase current, since the voltage range shall be limited by safety requirements for the electric vehicles. According to the phase current increasing, the design of electric machine windings shifts towards parallel paths, and, unfortunately, the skin effect and proximity effect tend to produce an uneven impedance of the parallel conductors, worsening additional Joule losses.

The design of the parallel paths requires a proper methodology to mitigate the additional losses and the current density distribution issues. The authors of [6] propose a method denoted as transposition, while other design methods have already been discussed and proposed to overcome these challenges [7].

This paper focuses on the reduction of the additional Joule losses through the variation in the geometric parameters of a single slot. The aim is to analyse if some slot parameters have a significant impact on additional Joule losses due to skin and proximity effects. This analysis considered both rectangular and trapezoidal slots. The analysis take into account three fundamental key performance indicators: total Joule losses, uneven current density distribution among slot conductors to avoid hot spots and uneven parallel path impedance to avoid unequal current-sharing in parallel conductors. The analysis considered a hairpin winding.

## 2. Analytical Approach

The current density distribution and the Joule losses can be analytically evaluated as a function of the operating frequency for a single rectangular slot without the polar shoes. Alternative hairpin winding layouts are taken into account in the model developed in [7].

The current density distribution can be analytically found following the Maxwell's equations, considering the reference frame of Figure 3 and assuming the magnetic field in quasi-static

approximation, see Equation (1). This approximation is reasonable, since the displacement current $\frac{\partial \vec{D}}{\partial t}$ for good conductors is negligible compared to the conduction (drift) current $\vec{J}$

$$\begin{cases} \nabla \times \vec{H} = \vec{J_z} \\ \nabla \times \vec{E} = -\frac{\partial \vec{B}}{\partial t} \end{cases} \tag{1}$$

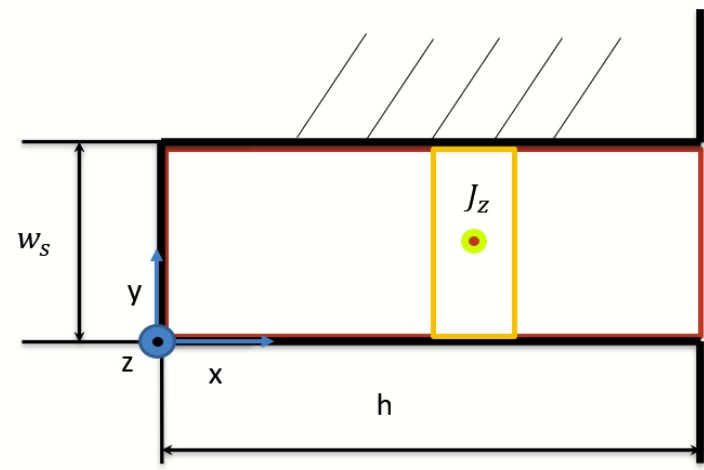

**Figure 3.** Single slot fully filled by a single conductor.

Merging the equations of system in Equation (1), the second-order partial differential equation for $J_z$ is obtained as in Equation (2), where $a$ is defined as in Equation (3).

$$\frac{\partial J_z^2}{\partial x^2} + \frac{\partial J_z^2}{\partial y^2} = a^2 J_z \tag{2}$$

$$a^2 = j\omega\mu\sigma \tag{3}$$

Considering the edges of the slot, due to the high difference between air and iron magnetic permeabilities, it becomes possible to consider only the $y$ components for the $\vec{B}$ and $\vec{H}$ fields. Therefore, the partial derivative of $J$ along the y-axis could be considered equal to zero, thus simplifying the expression for the $J_z$, as in Equtaion (4). The general solution of Equation (4) could be expressed as in Equation (5).

$$\frac{\partial J_z^2}{\partial x^2} = a^2 J_z \tag{4}$$

$$J_z(x) = c_1 e^{ax} + c_2 e^{-ax} \tag{5}$$

Applying the boundary conditions provided in Equation (6) to this equation, the mathematical expression for $J_z$ is obtained as shown in Equation (7).

$$\begin{cases} H_y(0) = 0 \\ I = \int \int \vec{J_z} \cdot d\vec{S} \end{cases} \tag{6}$$

$$J_z(x) = \frac{aI}{w_s} \frac{\cosh(ax)}{\sinh(ah)} \tag{7}$$

Substituting the expression for $J_z$ in the relation that defines the Joule losses, it is possible to obtain Equation (8), which represents the ratio between AC resistance and DC resistance, both per unit-length, where $\xi$ is defined as in Equation (9).

$$\varphi(\xi) = \frac{r_{AC}}{r_{DC}} = \xi \cdot \frac{\sinh(2\xi) + \sin(2\xi)}{\cosh(2\xi) + \cos(2\xi)} \tag{8}$$

$$\xi = \frac{h}{\delta} \tag{9}$$

Figure 4 shows the ratio between AC resistance and DC resistance as a function of $\xi$: the smaller the *skin depth* $\delta$, the greater the operating frequency and the AC resistance will be. In particular, $\varphi(\xi)$ grows linearly if $\xi \geq 2$ because the hyperbolic functions of Equation (8) become much greater than the trigonometric ones.

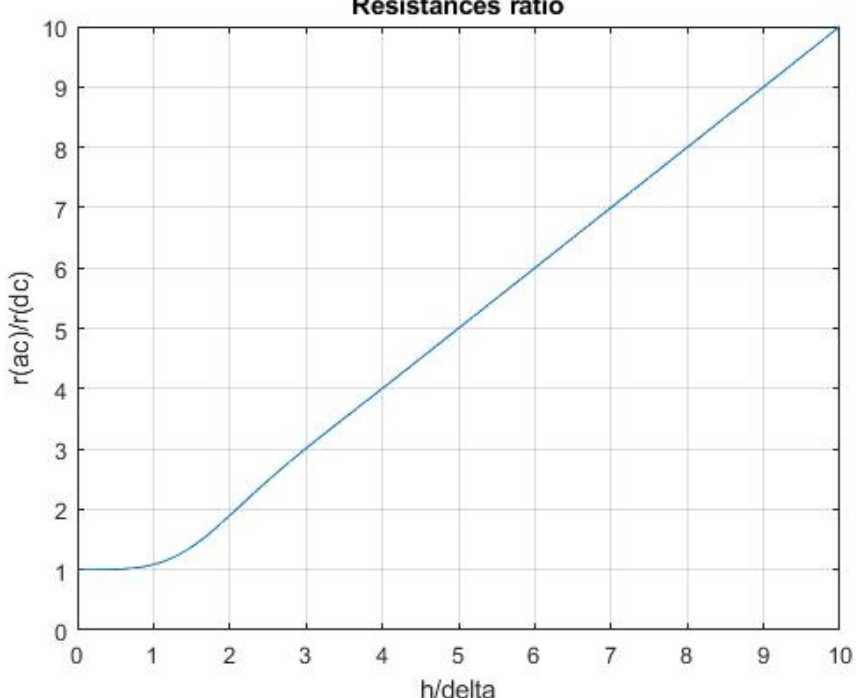

**Figure 4.** Ratio between $r_{AC}$ and $r_{DC}$.

This behavior is due to both skin and proximity effects, since the current density tends to distribute itself in a smaller surface than over the whole conductor, hence increasing the resistance. Since the current densities due to skin and proximity effects are orthogonal, as given in Equation (10), they can be treated separately, according to Chapter 6 and Appendix A of [8,9], respectively. Hence, the plots of the current density module and phase and the plot of the power losses can be found considering only one of the two effects. Chapters 3 and 4 of [10] treat both cases accurately.

$$\int\int (J_{z,s} \cdot J_{z,p}) \cdot dS = 0 \tag{10}$$

An analytical expression for the current density can be found, always starting from Maxwell's equation, Equation (1), when multiple conductors are placed inside the slot.

In this case, different boundary conditions have to be applied to the second-order partial differential Equation (2). These are given in Equation (11) and the final expression for the current density is obtained as in Equation (12).

$$
\begin{cases}
H_y\left(0\right) = 0 \\
\oint \vec{H}_y \cdot d\vec{l} = I_p \cdot \left(k - 1\right) \\
I_p = \int \int \vec{J}_z \cdot d\vec{S}_p
\end{cases}
\tag{11}
$$

$$
J_{z,k}\left(x\right) = a \frac{I_p}{w_p} \left\{ \frac{k \cosh\left(ax\right) - \left(k - 1\right) \cosh\left[a\left(x - h_p\right)\right]}{\sinh\left(ah_p\right)} \right\}
\tag{12}
$$

Equation (12) considers both skin and proximity effects, and it can also be found in [6]. The current density module along the conductors inside the slot is shown in Figure 5 at an operating frequency of 1000 Hz. This behavior [6] has also been validated by means of finite elements analyses (FEA).

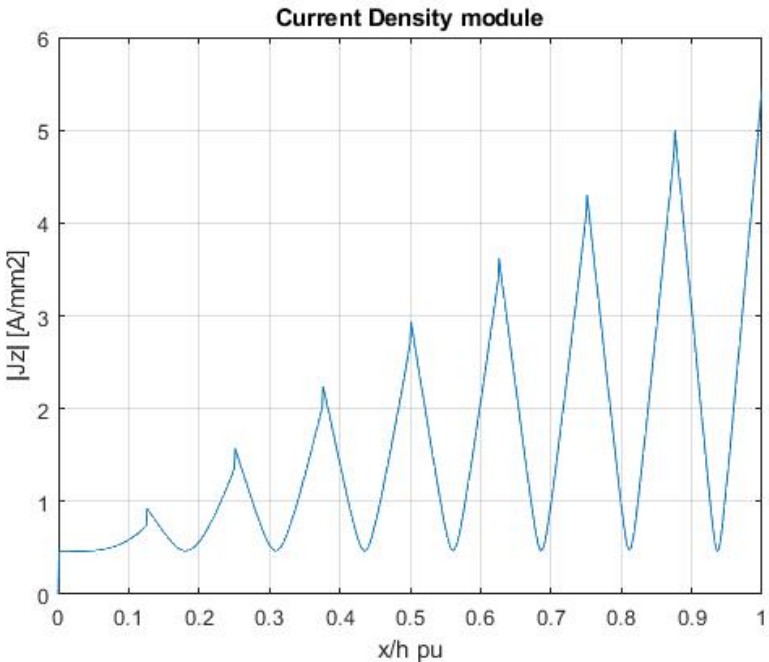

**Figure 5.** Current Density module for multiple conductors.

## 3. Finite Element Analyses

### 3.1. Introduction to the Analyses

Aiming at reducing the additional Joule losses potentially occurring in high-frequency operations, the relations between several slot parameters and the total loss distribution have been sought by the 2D FEA analysis. The clearance between conductors inside the slot has not been varied and the total MMF is kept at a constant value. The parameters under investigation are the stator slot opening $w_{so}$, the tooth width $w_t$ and the polar shoe thickness, see Figure 6. The FEA analysis is not intended to reduce the additional Joule losses through elimination of the closest conductor to the slot-opening, since this technique has been already studied in [11].

A single stator slot with four series-connected conductors was studied. Furthermore, two slot shapes were analysed: the rectangular and the trapezoidal shapes. Initially, the slot parameters were varied one by one, keeping the other constants. Then, two or three parameters were investigated

simultaneously: this is consistent with the multi-factorial analysis promoted by the *Design of Experiments* approach of [12].

The base dimensions in millimeters for the slot are shown in Figure 7.

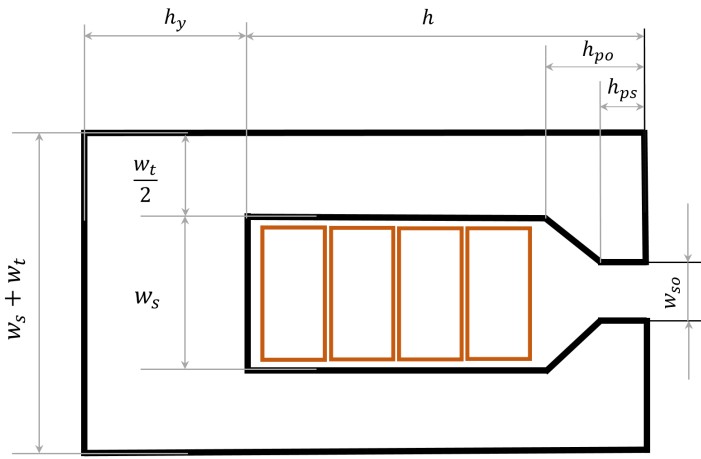

**Figure 6.** Stator slot parameters.

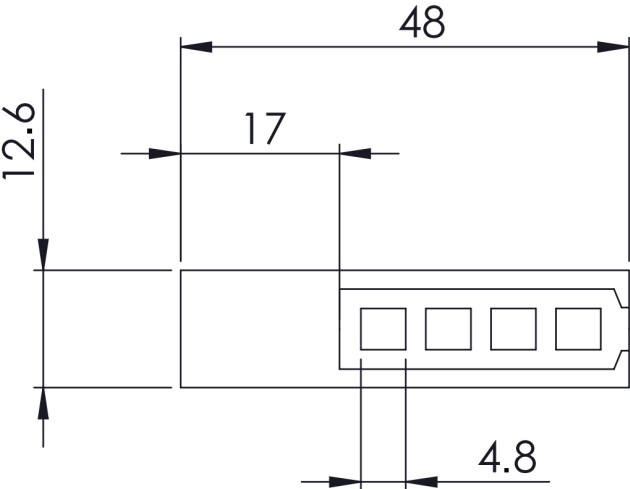

**Figure 7.** Rectangular slot, reference main dimensions in mm.

Two key performance indicators (KPIs) were employed to compare different simulation results:

- The percentage reduction, or the increment, in the total power losses;
- The percentage difference between the greatest and the smallest impedance $\Delta Z\%$ between the conductors inside the slot, see Equation (13).

$$\Delta Z\% = \frac{Z_{max} - Z_{min}}{Z_{min}} \tag{13}$$

The first KPI expresses which slot geometry leads to the lowest additional Joule losses, while the second KPI shows which configuration causes the slightest percentage impedance difference between all conductors inside the slot. The percentage difference impedance is of paramount importance in the

case of parallel conductors because if $\Delta Z\%$ is too high, the parallel paths will have an uneven current distribution. Since the BEV and the HEV usually work at high operating frequencies, the FEA analysis was carried out at 1000 Hz.

### 3.2. Slot Opening Width Analysis

### 3.2.1. Rectangular Slot

The slot-opening $w_{so}$ is the first analysed parameter. An array of slot-opening values was selected, and its extreme values corresponded to a narrow slot opening (see Figure 8b) and to the widest one (see Figure 8a), i.e., to a geometry without tooth shoe.

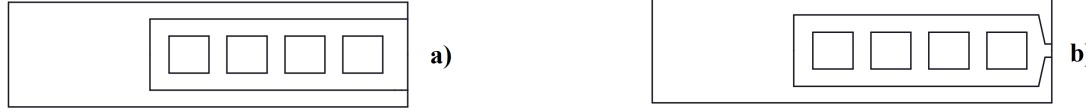

**Figure 8.** Rectangular slot with the widest (**a**) and the narrowest slot opening (**b**).

Varying the slot opening width does not significantly reduce, or increment, the additional losses for a rectangular slot, see Figure 9. In fact, considering the highest value of the total power losses (68.65 W) corresponding to the narrowest slot-opening, the percentage reduction corresponding to the widest slot-opening is 4.52%. On the other hand, $\Delta Z\%$ passes from 40.39% to 83.56%, ranging from the narrowest to the widest slot-opening; see Figure 10.

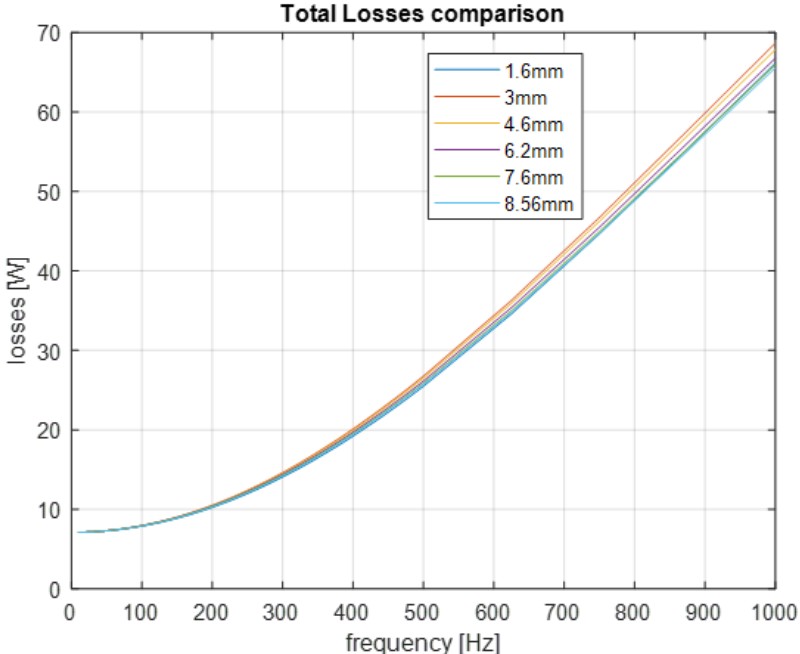

**Figure 9.** Rectangular slot—Total Losses comparison.

FEA results on slot-opening variation are antithetical, because the considered KPIs lead to different design choices.

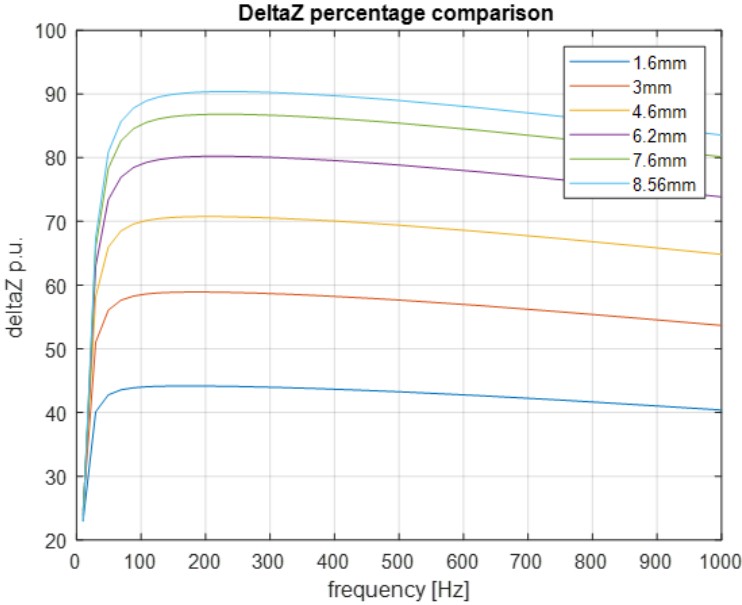

**Figure 10.** Rectangular slot–impedance comparison.

### 3.2.2. Trapezoidal Slot

The same slot-opening analysis was carried out for the trapezoidal shape and the FEA results lead to discordant results. A narrow slot opening, as in Figure 11, guarantees significantly lower total power losses and a smaller percentage difference between impedance, $\Delta Z\%$. More specifically, the total power losses percentage reduction corresponds to 15.64% if the narrowest slot-opening is chosen rather than the widest one, see Figure 12, and $\Delta Z\%$ equals 56.49% instead of 85.16%, see Figure 13.

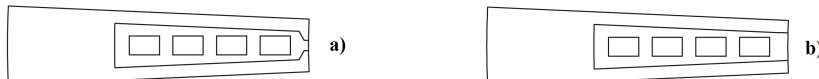

**Figure 11.** Trapezoidal slot with the narrowest (**a**) and the widest slot opening (**b**).

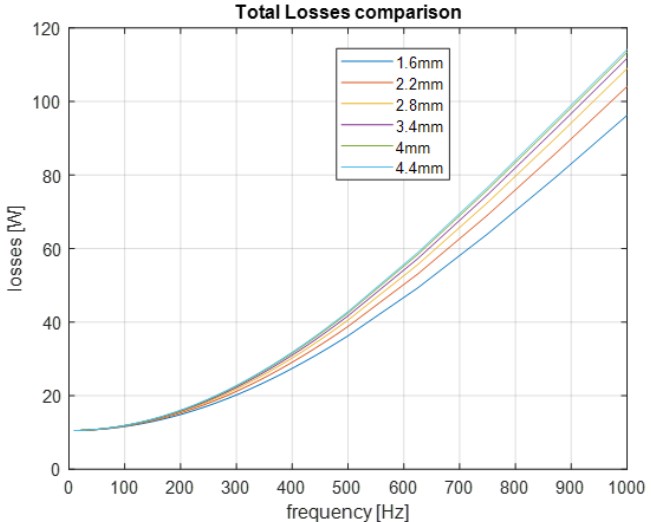

**Figure 12.** Trapezoidal slot—total losses comparison.

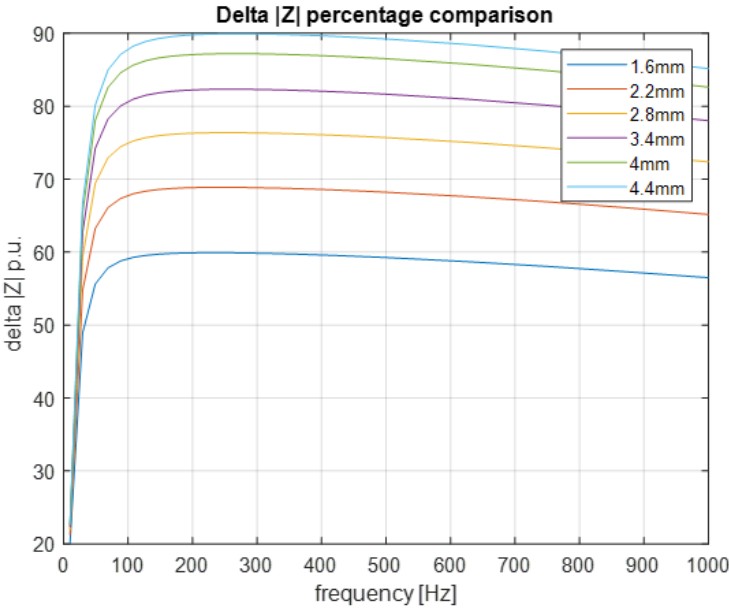

**Figure 13.** Trapezoidal slot—impedance comparison.

This inconsistency implies that another design parameter, that makes the slot-opening width significant, could exist: the teeth width $w_t$. In fact, if the teeth are too wide, the ferromagnetic material does not reach the saturation state and the magnetic flux paths are not efficiently employed.

*3.3. Tooth Width Analysis*

Trapezoidal Slot

In this analysis, the slot area, the filling factor and the pole shoes are kept constant and the *MMF* is the same for each simulation. The colour plots in Figures 14 and 15 show the influence of the tooth width, as claimed in the previous paragraph: a wide tooth reduces the $\vec{B}$ field induced inside the stator teeth. Furthermore, the ferromagnetic material does not even saturate with a wide tooth geometry, see Figure 15. The maximum value reached by the flux density is lower than 1.7 *T* if the tooth width, $w_t$ is greater than 3 mm, see Figure 16.

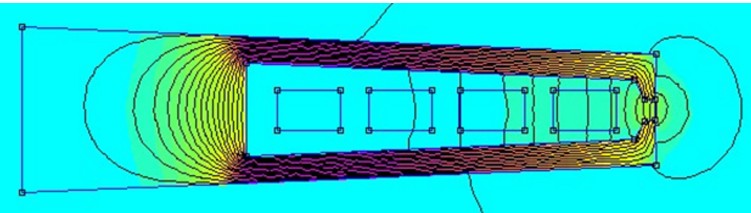

**Figure 14.** Trapezoidal slot with narrow teeth.

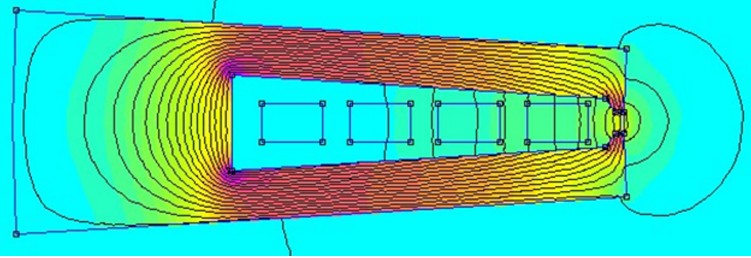

**Figure 15.** Trapezoidal slot with wide teeth.

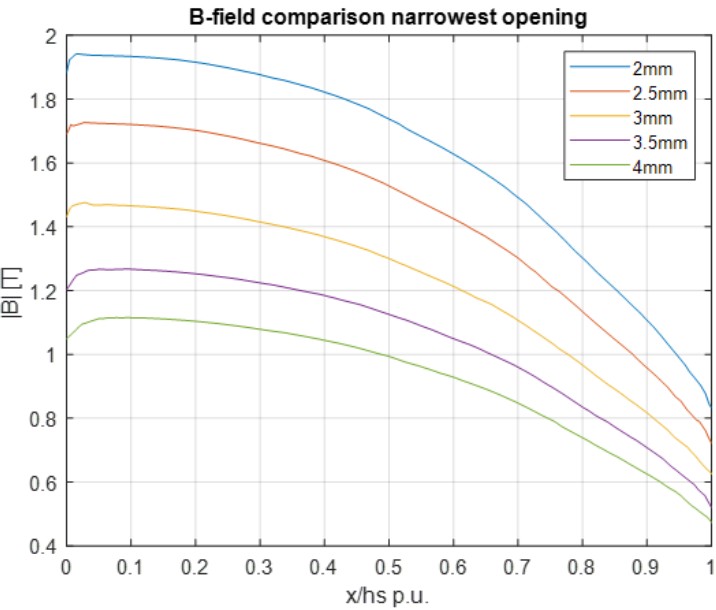

**Figure 16.** B-field comparison for a trapezoidal slot.

Considering the slot geometry of Figure 14, it is important to maintain a narrow slot-opening, such as 1.6 mm, since it leads to a percentage total power losses reduction of 15.69%, see Figure 17. This value is reduced by 0.31% when the tooth width is equal to 4 mm. On the other hand, the difference between $\Delta Z\%$ values for different slot-opening widths, is still significant and independent of the tooth width. According to the FEA results, the lowest percentage difference between impedance can be obtained with a geometry with a slot-opening of 1.6 m and a tooth width of 2 mm: $\Delta Z\% = 56.49\%$, see Figure 18.

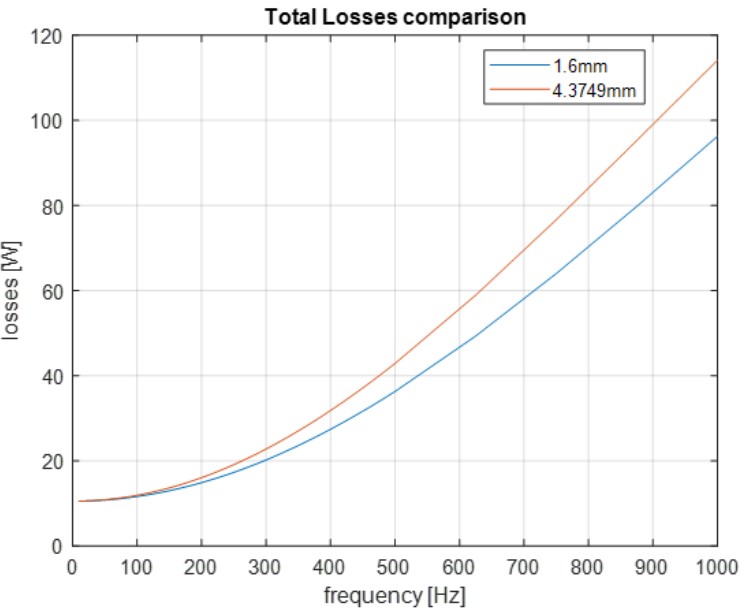

**Figure 17.** Slot opening total losses comparison, wt = 2 mm.

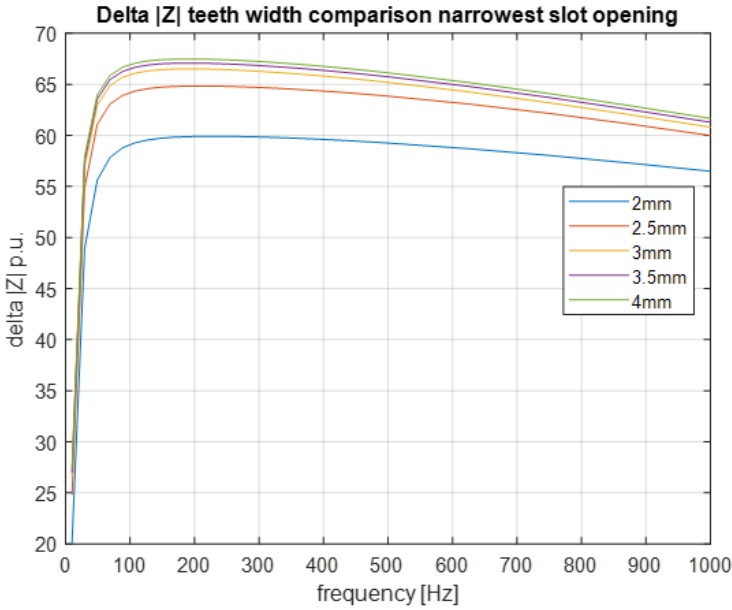

**Figure 18.** impedance comparison for different teeth width, wso = 1.6 mm.

It is possible to replicate this analysis for the rectangular shape, following the same considerations.

### 3.4. Polar Shoe Depths Analysis

Trapezoidal Slot

Another analysis concerns the variation in both polar shoe depths, i.e., the variation in the two bases of each trapezoid, that shapes each polar shoe. Both depths were varied simultaneously and a ratio of 1:2 was constant between them.

Considering a small polar shoe as the slot in Figure 19, the ferromagnetic magnetic material saturates inside the polar shoes even if the flux density does not reach high values as inside the stator tooth. On the other hand, if the polar shoe depths is high, as in Figure 20 ($h_{ps} = 4$ mm and $h_{po} = 2$ mm), the polar shoe saturation does not occur. Hence, a smaller $\vec{B}$ inside the polar shoes means a smaller $\vec{H}$ on the closest conductor to the slot-opening, and thus lower total power losses associated to it. As well as in the previous tooth width analysis, two slot-opening widths were considered: the narrowest of 1.6 mm and the widest of 4.4 mm.

The gap between the *Total Losses* becomes wider: this is verifiable by comparing Figures 17 and 21. In fact, the *Total Losses* percentage reduction passes from 15.69% to 25.79% and the lowest value, reached at 1000 Hz, drops to 71.44 W. To summarize, the additional losses go from 96.28 W to 71.44 W, a further percentage reduction of 25.8%.

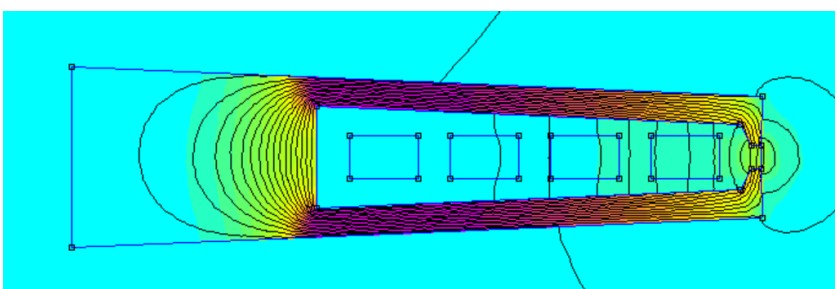

**Figure 19.** Polar shoes with the smallest depths.

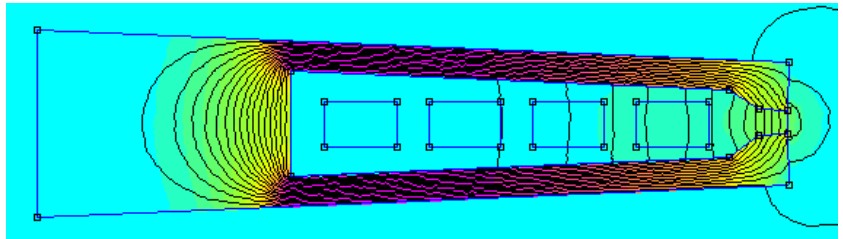

**Figure 20.** Polar shoes with the greatest depths.

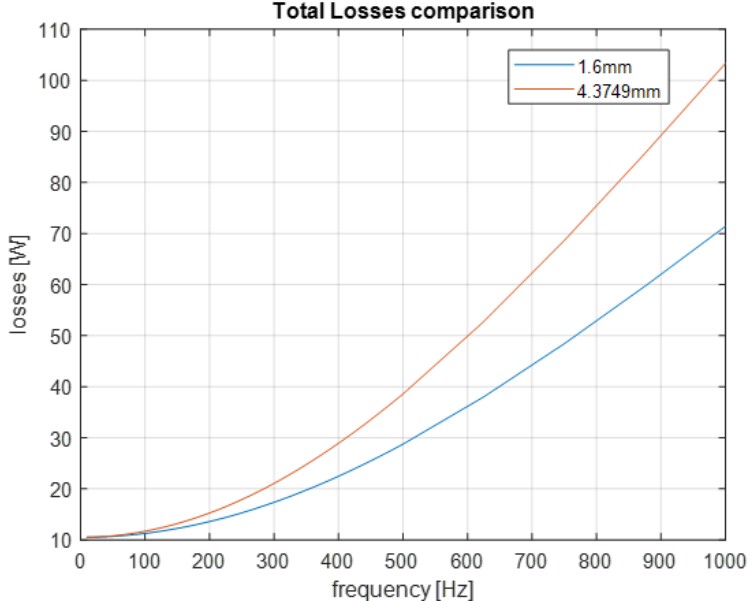

**Figure 21.** Slot opening total losses comparison, hps = 4 mm.

As well as for the previous analysis, the gap between ΔZ% values at different openings stays wide, and its lowest value is 37.61 % Figure 22, which corresponds to the largest depths, against 56.49% which corresponds to the smallest ones.

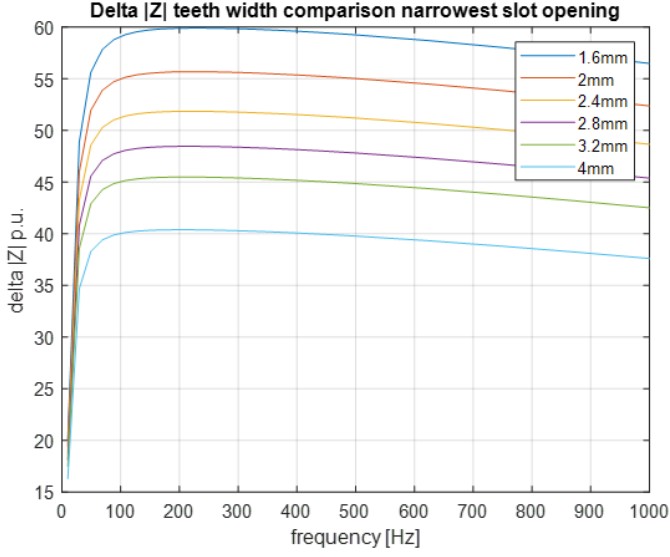

**Figure 22.** impedance comparison for different depths.

Even for this analysis, the same results would have been obtained simulating the rectangular shape.

*3.5. Slot Depth to Width Ratio Analysis*

The final analysis is related to the ratio between slot depth and slot width. This analysis is based upon the following assumption: smaller conductors could mitigate the skin effect. In order to effectively carry out this analysis, it is necessary to keep the slot area, $S_{slot}$, constant, as well as the filling factor, $k_{fill}$, to have the same quantity of copper inside the slot and, thus, the current applied to each conductor. An RMS current density of 10 A/mm$^2$ was imposed and only the rectangular shape were considered.

Rectangular Slot

The geometry simulated is characterized by wide tooth, thus the widest slot opening guarantees the minimal additional losses will be obtained coherently with the tooth width analysis. The slot depth to slot width ratios chosen were: 1.5, 2, 2.5 and 3. The percentage total power losses reduction between the ratio $R = 3$ and the ratio $R = 1.5$ is very important: 62.67%, from 51.12 W to 19.07 W, Figure 23. Furthermore, a ratio of 1.5 guarantees a low $\Delta Z$%: 14.94%, if the narrowest opening is chosen. As well as for the slot opening width analysis, a narrow opening leads to a smaller difference between the impedance.

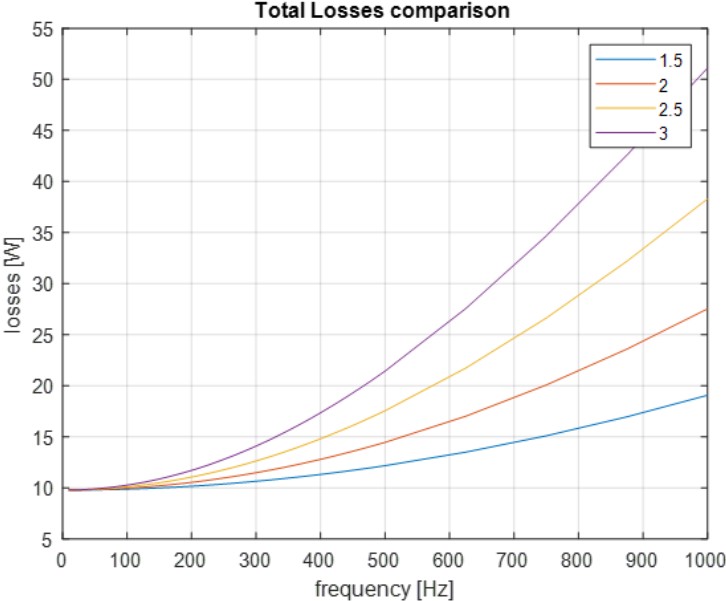

**Figure 23.** Total losses comparison—different ratios.

Even the current density distribution is enhanced: in fact, even if it remains uneven, especially for high operating frequencies, the average current density value inside the conductors gets closer to the maximal one, see Figure 24. That is assumable from the colour plot as well; the "best case" is presented in Figure 25.

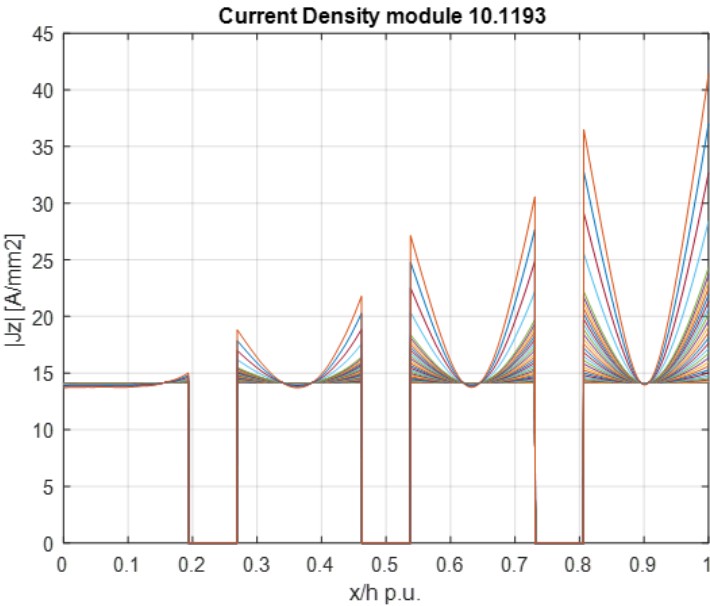

**Figure 24.** Current Density module—widest opening.

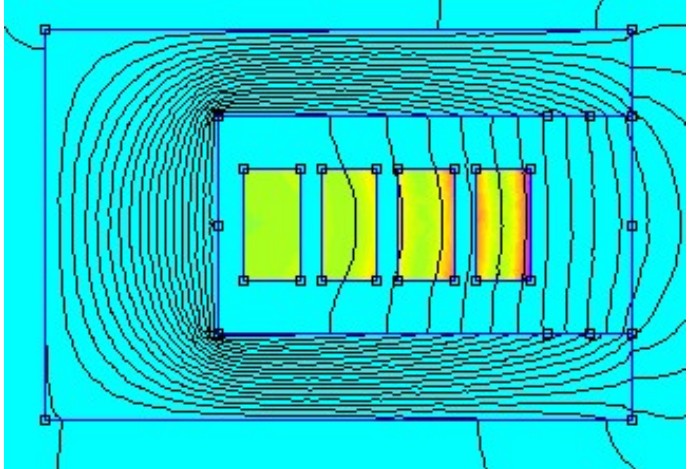

**Figure 25.** Colour plot of the Current Density.

## 4. Conclusions

The analysis results show how some stator slot parameters can have a meaningful impact on the additional Joule losses occurring in hairpin windings. The considered slot parameters are: the slot-opening width, the stator tooth width, the polar shoe depths and the ratio between slot depth and slot width. In particular, the FEA results suggest that a proper design of the stator slot would include the following details:

- Narrow stator teeth;
- Great polar shoes, in order to create the best flux density paths;
- Narrow slot-opening.

Another relevant parameter is the slot depth to slot width ratio; in particular, it should feature a value of around 1.5.

If the previous design's expedients are accurately combined, the slot design could be a good way to mitigate the consequences of skin and proximity effects on additional Joule losses. Figure 26 shows a final slot geometry for the rectangular shape that takes into account all of the design recommendations of this paper.

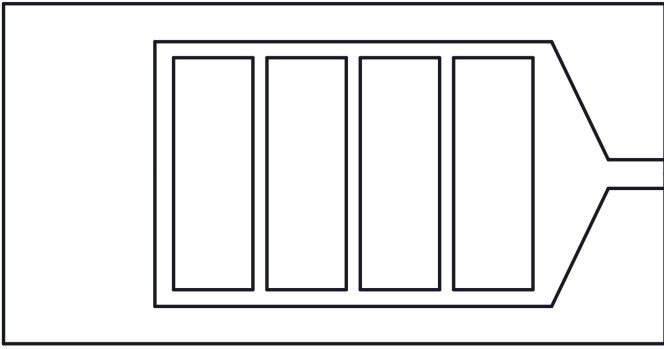

**Figure 26.** Rectangular slot—best geometry.

The geometry in Figure 26 also reduces the percentage difference between the impedance with the highest impedance conductor and the lowest impedance conductor, ΔZ%. Figure 27 shows the ΔZ% value of 6.23% at 100 Hz frequency, obtained with the optimal rectangular slot geometry. Because of the small difference, adopting parallel paths should be less burdensome for this slot geometry.

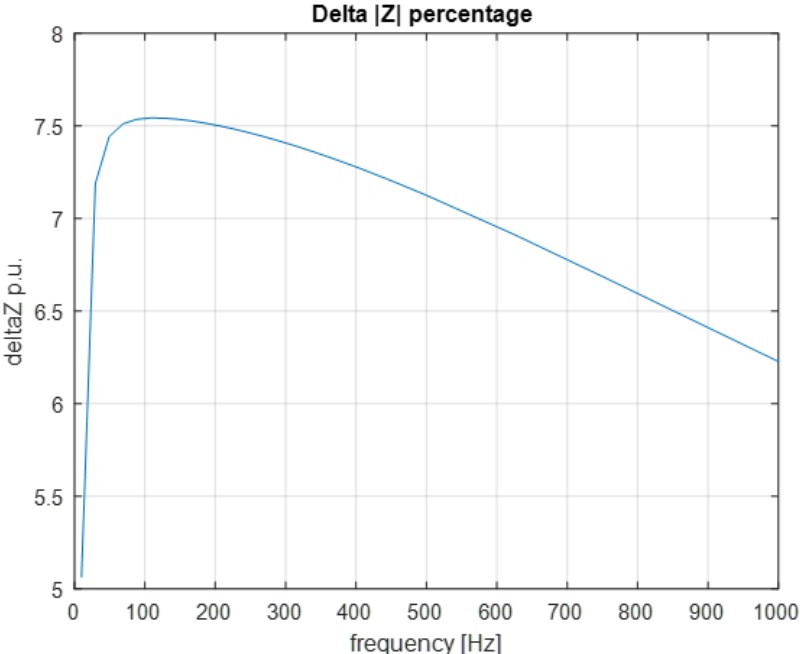

**Figure 27.** impedance comparison.

**Author Contributions:** Conceptualization and methodology, C.B. and M.V.; validation, S.N. and D.B., investigation, C.B., M.V. and A.T.; data curation, A.T.; writing—original draft preparation, M.V.; writing—review and editing and supervision, G.F. All authors have read and agreed to the published version of the manuscript.

**Funding:** This research received no external funding.

**Conflicts of Interest:** The authors declare no conflict of interest.



## Abbreviations

The following abbreviations are used in this manuscript:

BEV     Battery Electric Vehicles
HEV     Hybrid Electric Vehicles
FEA     Finite Element Analysis
KPI     Key Performance Indicator

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
