# Peer review of "Slot Design Optimization for Copper Losses Reduction in Electric Machines for High Speed Applications"

_applsci, doi:10.3390/app10217425_

Round 1

Reviewer 1 Report

This work aims to study the effect of slot design on the AC copper loss (resistance increase due to skin and proximity effect). In the introduction, the background and motivation of this study is clearly demonstrated. The study using the FEM identifies the critical parameters in slot geometry that affect the AC resistance. 

  • It is recommended to mark (describe) the geometric parameters in the figure. For example, where exactly is the “opening width” located in Figure 6, and “teeth width” in Fig. 12, and so on.
  • AC resistance might be affected by the current strength. Because, the soft ferromagnetic material in the slot has nonlinear B-H curve that is determined by the current. Thus, it is recommended to comment about the effect of current strength, although this work focused on the effect of the frequency.
  • In Fig 23, where is mu-metal located? Is the whole slot part composed of mu-metal? I guess that coil is surrounded by mu-metal “foil” to avoid the penetration of magnetic field into the coil. Please clarify the configuration in Fig. 23.

Reviewer 2 Report

This paper reports a method for slot design optimization to reduce copper losses in electrical machine for high speed applications. Such power losses are mainly due to the eddy currents, which are manifested as power losses due to skin and proximity effects.

Although the idea is good, since it falls in a very interesting research area, the work has some important points to improve as detailed below.

1.- English grammar and style require to be revised all along the manuscript.

2.- Manuscript title. It must include the “electric machines” or “electric motors” or even the type of electric machines in which the work focuses.

3.- Introduction section. It must clearly describe the main novelties and achievements of the paper with respect to the state of the art, and the advantages and drawbacks (if any) of the proposal with respect to similar solutions and developments.

4.- It seems that (1) includes the quasi-static approximation. Please develop.

5.- Eq. (3) and Eq. (5) need a reference.

6.- Please explain the meaning of h and delta in Eq. (6).

7.- Page 5. “Thist trend has also been experimentally obtained in the finite elements analyses (FEA).” Please add a reference

8.- Page 6. Please explain the meaning of delta(Z)% and its importance

9.- The section devoted to the mu-metal is irrelevant

10.- In general graphs quality must be improved

11.- The reference list must be updated. More references are required, specifically from 2018-2020.

The Reviewer aims the authors to revise the work based on the suggestions above in order to improve its quality.

Round 2

Reviewer 2 Report

The authors have replied my concerns